# Relationship between Viewing Motivation, Presence, Viewing Satisfaction, and Attitude toward Tourism Destinations Based on TV Travel Reality Variety Programs

**Bo-Kyeong Kim ***  **and Kyoung-Ok Kim**

Department of Tourism Management, Pukyong National University; 45 Yongso-ro, Busan 608737, Korea; okson334@hanmail.net
* Correspondence: ynnij@hanmail.net

**Abstract:** The purpose of this study was to clarify the relationship between viewing motivation for reality programs and the viewing experience (presence) during watching, viewing satisfaction after watching, and attitudes toward presented tourism destinations. While this study notes that the empirical verification of travel reality variety programs is insufficient, various discussions are presented with regard to the grip of reality variety program fever in Korea. Notably, viewers are interested in the emotional experience related to characters and tourism destinations in reality variety programs. Therefore, we asked the following questions: (1) What kind of motivation encourages viewers to watch reality variety programs? (2) How does motivation for viewing a reality variety program affect viewing satisfaction through a certain approach (presence)? (3) How does viewing satisfaction affect one's attitude toward a program's tourism destinations? To answer these questions, we conducted a survey of 358 viewers of travel reality variety programs. The results of this study are as follows: (1) viewing motivation for travel reality variety programs consists of five factors: vicarious gratification, entertainment, information-seeking, habitual time-spending, and socializing; (2) it was confirmed that the effect of these five factors on satisfaction was mediated by presence (viewing experience); and (3) viewing satisfaction through presence affected the subsequent attitude toward presented tourism destinations.

**Keywords:** travel reality variety program; viewing motivation; viewing satisfaction; presence; attitude toward tourism destination

## 1. Introduction

In the past, most programs about travel were mainly documentaries focused on information delivery. However, the recent trend involves reality variety programs, which combine variety, fiction, and reality; travel programs combine entertainment and seeking fun, transforming into a new genre called "travel entertainment" [1]. It is said that this change in travel programs is leading to corresponding changes in travel pursuit and tourism trends [2]. In addition, after travel reality variety programs are broadcast, credit card usage for travel and leisure increases. This suggests that consumers' or viewers' desire to travel is indirectly connected to travel programs.

Reality variety programs are continuously increasing in terms of attractive content due to accelerated competition among media, as well as the advantage of securing viewer ratings at relatively low production costs [3–6].

Due to the format of reality travel programs, tourist destinations are shown on the screen frequently and for longer periods of time compared to drama programs and movies. Therefore, viewers are more

likely to be more immersed in the presented situation than they would be with dramas and movies, They also form images of tourist destinations in the programs and make meaningful connections to the selection of a travel destination [7].

As such, mass media plays an important role in the travel industry, and the reality variety format is particularly important for theoretical research and has managerial aspects that can be key in the travel industry.

In particular, travel programs in the reality format are increasing and the formats are diversifying, but related previous studies are limited to the exposure effects on the tourist destinations, factors influencing the choice of destinations, and airline ticket price analysis for travel programs.

Recently, there have been several studies on how viewing motivation for reality travel programs influences future viewer behavior. However, they are limited to discussing the general relationship between viewing motivation and viewing satisfaction.

Therefore, this study aimed to investigate the effects of viewing motivation, viewing satisfaction, and attitudes toward tourism destinations, and to empirically reveal the mechanism of presence (watching experience) in a sample of men and women over the age of 18 who live in Busan and Gyeongnam provinces. To achieve this purpose, based on previous research on viewing motivation for broadcast programs, it is classified into five factors related to reality travel programs, and the examination of the effect of the five factors on satisfaction is mediated by presence (viewing experience). Lastly, viewing satisfaction through presence is confirmed with regard to its influence on subsequent attitudes toward presented tourism destinations.

## 2. Literature Review

### 2.1. Viewing Motivation for Travel Reality Variety Programs

Previous studies on viewing motivation divided it into social reward and psychological motivation [8], including information, entertainment, establishing a relationship, and recreation [9]; and entertainment, time-spending, and social learning [10,11]. Viewing motivation is based on uses and gratification theory [12,13], which posits that viewers use or watch TV programs to satisfy their desires and needs. Unlike traditional research about the effect of mass media asking questions such as, What does media do to people? this theory conversely asks, What do people do with media? Thus, this theory is used to understand viewers' media experience and to identify their motivation in selecting and using specific media in comparison with other media.

Research on uses and gratification has been extended to studies ranging from categorizing viewing motivation for TV programs to describing viewer motivation according to the genre of programs.

According to Lim (2008) [14], college students' motivations for viewing American dramas were entertainment and rest, information, and environment/companion, while those for Korean dramas also included drama characteristics (feature) and habitual free time-spending.

Ban and Park (2014) [15] studied viewing motivation for reality dating programs and categorized it into virtual avoidance, indirect experience, pleasure, social interaction, and free time-spending. For a certain type of reality travel program, such as "Daddy! Where Are You Going?", there are four main factors of viewing motivation: interest, the attractiveness of the children, alleviating loneliness, and expressing everyday emotions [16,17]. "Hyori's B & B," which aired in 2017 and 2018, was an observational reality travel program that recently became an entertainment trend, with entertainment, empathy, interaction, aesthetics, information, and indirect experiences selected as viewing motivations [2]. It was found that only aesthetics, information, and indirect experiences had significant effects on viewer satisfaction and intention to visit.

The results of these studies show differences in viewing motivation according to program format or genre, and that generally, but not always, it affects viewer satisfaction.

Therefore, the motivation for viewing travel reality variety programs does not only involve acquiring information about tourism destinations, but there is also a psychological motivation in

comparison with other travel programs [16]. Accordingly, this study proposes vicarious gratification, entertainment, information-seeking, habitual time-spending, and socializing as factors of viewing motivation for travel reality variety programs based on previous studies.

## *2.2. Presence*

While the definition of presence varies according to the researcher [18], it generally refers to the psychological and subjective experience of a person's feeling of being directly in the mediating environment outside their surrounding physical environment [19–21]. To be specific, there are six subtypes of this dimensionality: (1) social richness that feels the media intimately, (2) realism based on realistic expression, (3) transportation that seems to be going to a virtual media environment, (4) immersion in a virtual reality, (5) a sense of reality as a social actor within the medium who interacts with the mediated character, and (6) feelings toward the medium itself as a social actor [20]. The study of this type of presence has been extended to its role as a mediator between viewing motivation and satisfaction, as well as various media environments. According to Kim and Biocca (1997) [19] and Lombard and Ditton(1997) [22], while there are differences in the types and scopes of media, presence can be applicable to all media. In addition, presence is established by the format, genre, content, and characteristics of the media, and can be sufficiently experienced even in a traditional medium where the mediated environment is relatively unrealistic. In particular, in the case of TV media, the amount and precision of sensory channels mobilized by users, i.e., the "sensory output", creates a higher reality experience that is called presence in TV but not radio, which is only auditory [23].

Therefore, it can be said that it is difficult to explain the genre of reality travel programs only by the relationship between viewing motivation and satisfaction in terms of the realistic expression (presence as reality) of TV media [8,24]. In particular, in the case of reality travel programs, with travel and storytelling, viewers become more immersed in the tourism destination, which is the background of the program, making them more aware of the reality of the place depending on the level of perceived similarity and wishful identification with the characters [1]. In accordance with this view, it is critical to confirm that presence plays a mediating role in the relationship between viewing motivation and satisfaction [25].

Based on the above discussion, this paper examines the mediating role of presence in accounting for the relationship between viewing motivation and viewing satisfaction for travel reality variety programs. We accordingly set out to prove the significance of presence as a parameter that affects viewing satisfaction.

## *2.3. Viewing Satisfaction*

Satisfaction with viewing media and programs starts with expectations for content. Thus, since satisfaction with a program is shaped when the viewer's expectations for that program are met, or the viewer evaluates them positively through a process of assessing the cognitive and emotional dimensions of the content, their consequent satisfaction reflects the fulfillment of their desires. Therefore, satisfaction after watching a program is a very critical parameter for understanding the viewer's media experience. Satisfaction with television viewing is formed by a combination of various factors, such as motivation and viewing experience. In fact, there has been extensive research on viewing satisfaction and viewing experience regarding television and specific programs [11,24,26].

According to the expectancy value model of Palmgreen and Rayburn (1984) [27], media consumption is driven by satisfaction, or the seeking of gratification, and perceived gratification obtained afterwards. Beliefs and evaluations are modified, thereby affecting the repeated seeking of gratification. In other words, if gratification sought and obtained is consistent, viewing satisfaction increases, which leads to continued viewing. This model explains the cyclical relationship between viewing motivation, viewing behavior, and satisfaction (fulfillment) through various media.

Indeed, on travel reality programs, tourism destinations are realistically described from the tourists' perspective, and viewers' expectations and desires are satisfied by being providing rich

information on tourism destinations [10,28]. In addition, detailed information about the region/place is provided while famous tourists travel directly in the program, unlike in drama and movies [6,29]. In short, as it is a place that is not merely an image, such as in a film or drama, but is directly exposed to viewers, it can be expected to arouse their expectations and intentions for future behavior.

Therefore, this study intends to expand viewing satisfaction theoretically through the specific parameter that the motivation to view travel reality variety programs affects viewing satisfaction, based on previous studies.

## 2.4. Attitude toward Tourism Destinations

Attitude is an ongoing assessment of beliefs and emotions about an object [30,31]. Attitudes are also individuals' thoughts, emotions, and tendencies regarding particular subjects, and their future behavior will vary depending on how they establish attitudes toward particular subjects [32]. In addition, it is not necessary for awareness, influence, and behavior to simultaneously exist in order to form or express an attitude; rather, such can be established when one or more components are combined [33]. Fishbein and Ajzen (1975) [31] proposed the theory of reasoned action by adding subjective norms to attitudes and behavioral intentions to clarify the human decision-making process. Ajzen (1991) [34] then developed the theory of extended planned behavior by adding perceived behavior control considering the external environment. In the tourism industry, it is important to understand tourists' decision-making, so research has been performed to identify behavioral intent through attitude [28,35–37]. In particular, decision-makers include many criteria when selecting tourism destinations, determining their preferences and behaviors concerning future destinations by comparing and evaluating the attributes of tourist sites [38].

Thus, tourists can be said to have different behaviors depending on their prior preferences [39]. Tourists are exposed to diverse mass media, actively use media to gather information about tourism destinations, and make various decisions regarding tourism, such as selecting destinations, through such information. The media sector that has attracted the most attention recently in the field of tourism is entertainment, which is indirectly connected to viewers (potential tourists) through movies and TV programs, and directly connected to the tourism industry [40]. Based on previous research, it is necessary to extend the research area into the effect of viewing satisfaction on attitudes toward tourist destinations according to the motivation for viewing travel reality variety programs.

## 3. Research Model and Methodology

### 3.1. Research Model and Hypotheses Development

Viewing motivation is a significant factor in understanding how to meet viewer expectations for a program as well as influencing attitudes and future behaviors. Taking a look at previous studies, there was a difference in viewing motivation depending on the viewer or the program format or genre. In particular, the effect on viewing satisfaction or other behavioral variables can be different through the realistic expression (presence as reality) of the place shown in a travel reality program, unlike in films and dramas [7,40,41]. Therefore, presence is a psychological factor of the viewer and can be said to be a weighty parameter that influences satisfaction with the use of TV or media [5]. For example, viewers motivated to pursue information on travel will have a higher level of interest in tourism destinations that are in the background of the program. Particularly, while moving along with performers, they can experience presence in the places they see on the screen, and this shows high satisfaction with the program [15].

Therefore, the following hypothesis was proposed:

**Hypothesis 1 (H1).** *Viewing motivation for a travel reality variety program positively affects presence.*

**Hypothesis 1-1 (H1-1).** *Vicarious gratification positively affects presence.*

**Hypothesis 1-2 (H1-2).** *Entertainment positively affects presence.*

**Hypothesis 1-3 (H1-3).** *Information-seeking positively affects presence.*

**Hypothesis 1-4 (H1-4).** *Habitual time-spending positively affects presence.*

**Hypothesis 1-5 (H1-5).** *Socializing positively affects presence.*

Previous studies on viewing motivation and satisfaction found a positive relationship between these factors through the mediation of parasocial interaction and presence on the program "Cookbang" [15]. This means that high levels of involvement in, attention to, and engagement with the reality program by viewers showed a positive effect on their viewing satisfaction according to their watching experience (presence). Therefore, this reality presence can increase enjoyment and engagement even when watching a travel reality variety program, which can increase viewing satisfaction. Hence, the following hypothesis was proposed:

**Hypothesis 2 (H2).** *Presence positively affects viewing satisfaction with a travel reality variety program.*

Viewing satisfaction means that a program exceeds the viewer's expectations, and viewers are more satisfied with a program if their expectations are exceeded, and dissatisfied if their expectations are not met [11]. Satisfaction with TV viewing is influenced by the viewer's attitude and behavioral intention, because the viewing motivation or expectation for the program is satisfied or the viewer's behavior is closely related to the program [42]. Therefore, the following hypothesis was proposed:

**Hypothesis 3 (H3).** *Viewing satisfaction with a travel reality variety program positively affects the viewer's attitude toward the tourism destination.*

Based on the above hypotheses, this study's research model is shown in Figure 1.

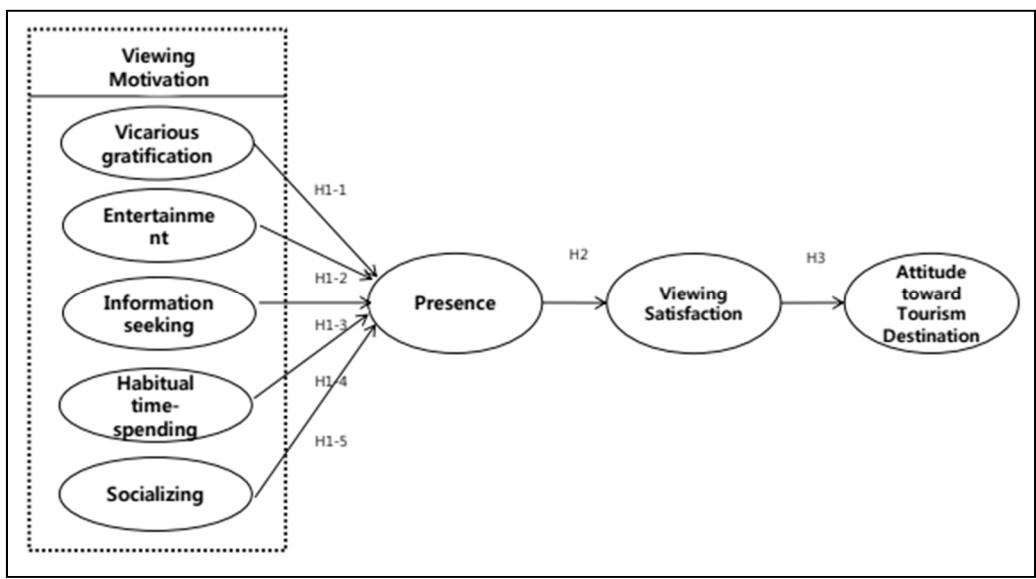

**Figure 1.** Research model.

*3.2. Operational Definition*

Viewing motivation was defined as "the viewer's personal viewing desire/need for the media" [8]. Next, based on the research of McQuail (2005) [36], Ebersole and Woods (2007) [43], and Jang and Kim (2016) [44], five motivational factors were derived: vicarious gratification, entertainment,

information-seeking, habitual time-spending, and socializing. With slight modification, a seven-point Likert scale was used to measure these factors (1 = strongly disagree to 7 = strongly agree). Presence was defined as "feelings/emotions of immersing into the environment created by television programs without being able to perceive the reality of television" based on the research of Bracken (2009) [23] and Pope and Wansink (2015) [11]. With slight modification, a seven-point Likert scale was used to measure four items (1 = strongly disagree to 7 = strongly agree). Viewing satisfaction was defined as follows: "the viewer uses media to satisfy his or her various needs/desire and establish satisfaction," from the uses and gratification theory of Kartz (1959) [9]. Four items were extracted and slightly modified from Kim and Rubin (1997) [5] and An and Han (2018) [16], and measured by a seven-point Likert scale (1 = strongly disagree to 7 = strongly agree). Lastly, attitudes toward tourism destinations were measured by using a seven-point Likert scale (1 = strongly disagree to 7 = strongly agree) for the four items used in the survey on the impact of the series "Better Late Than Never" of the Korea Tourism Organization-International Travel Trend Report (2014).

### 3.3. Data Collection and Analysis Method

For this study, the survey was conducted via an Internet questionnaire, first confirming the experience of watching reality variety programs. Next, the authors set as samples those living in Busan and Gyeongnam provinces who had experience watching travel reality variety programs from 1–31 March 2020. The convenience sampling method was applied. Among the 382 total respondents, 358 valid questionnaires were received and used for analysis, excluding (1) nonresponse, (2) overlapping response, and (3) unfinished questionnaires. Because of the high response rate, analysis of nonrespondents was not conducted, since a 94.7% response rate was considered adequate [45]. SPSS 25.0 and AMOS 23.0 were used for the analysis.

The statistical analysis methods and procedures applied in the empirical data analysis process of this study were as follows:

First, frequency analysis was performed to examine the general demographic characteristics of the sample. Second, exploratory factor analysis was conducted to identify the factor structure, reliability, and validity of the measurement tool, and confirmatory factor analysis and discriminant validity analysis were performed to verify the fit of the factor structure. Third, Structural Equations Models (SEMs) were examined to investigate the hypotheses.

## 4. Results

### 4.1. Profile of the Sample

The demographics of the 358 participants were as follows: 188 men (52.7%) and 170 women (47.3%). Regarding the age groups, 103 people (29.0%) were in their 20s, 113 people (31.7%) were in their 30s, 88 people (24.6%) were in their 40s, and 38 people (10.7%) were in their 50s. In terms of marital status, 230 (64.5%) were married and 128 (35.5%) were single. Regarding the distribution of occupations, 51 were homemakers (14.3%), 84 had specialized jobs (23.7%), 193 (54.0%) were office workers, and 30 (8.0%) had other occupations. Regarding the distribution of the amount of travel in a year, 119 people (33.5%) traveled once, and 142 people (39.7%) traveled two to three times a year. Regarding the number of reality variety programs watched in a week, 54 people (14.7%) watched once per week, 177 people (49.6%) watched two to three times, and 76 people (21.4%) watched four to five times. Considering distribution of monthly income (in million won), 198 people (55.4%) earned 3 to 4.99 million won and 76 people (21.0%) earned more than 5 million won. The demographic characteristics of this study are shown in Table 1.

**Table 1.** Demographic variables (n = 358).

| Classification | | Frequency | Percentage (%) | Classification | | Frequency | Percentage (%) |
|---|---|---|---|---|---|---|---|
| Gender | Male | 188 | 52.7 | Amount of travel in a year | 1 time | 119 | 33.5 |
| | Female | 170 | 47.3 | | 2–3 times | 142 | 39.7 |
| Age (years) | 20 | 103 | 29.0 | | 4–5 times | 38 | 10.7 |
| | 30 | 113 | 31.7 | | 6–7 times | 17 | 4.9 |
| | 40 | 88 | 24.6 | | >8 times | 42 | 11.2 |
| | 50 | 38 | 10.7 | Number of reality variety programs watched in a week | 1 time | 54 | 14.7 |
| | >60 | 16 | 4.0 | | 2–3 times | 177 | 49.6 |
| Marital status | Married | 230 | 64.5 | | 4–5 times | 76 | 21.4 |
| | Single | 128 | 35.5 | | 6–7 times | 51 | 14.3 |
| Occupation | Homemaker | 51 | 14.3 | Monthly income (in million won) | <1.49 | 36 | 10.1 |
| | Specialized job | 84 | 23.7 | | 1.5–2.99 | 48 | 13.5 |
| | Office worker | 193 | 54.0 | | 3–4.99 | 198 | 55.4 |
| | Other | 30 | 8.0 | | >5 | 76 | 21.0 |
| Sum | | 358 | 100 | Sum | | 358 | 100 |

### 4.2. Reliability and Validity

This study confirmed the reliability and validity of the variables before hypothesis testing. Regarding the reliability of constructs, internal consistency was found with all measuring factors with Cronbach's α over 0.7 [46]. The validity of the variables was verified through confirmatory factor analysis. The results are presented in Table 2. First, it was concluded that the standardized factor loading value was over 0.6, the significance level was less than 0.1%, and the AVE (average variance extracted) value was over 0.5, indicating that there was convergent validity [47]. Comparing the AVE value with the square of the correlation coefficient (based on Table 3) showed that the AVE value of latent variables was greater than the square of correlation value. Thus, the validity of discriminant validity was assured [47].

**Table 2.** Results of confirmatory factor analysis of the measurement model.

| Construct | Measurement Items | Factor Loading | S.E. | C.R. | *p*-Value | Cronbach's a |
|---|---|---|---|---|---|---|
| Vicarious gratification | While watching reality travel variety programs, I feel assimilated with the characters. | 0.728 | | | | 0.902 |
| | While watching reality travel variety programs, I can forget my daily life. | 0.755 | 0.074 | 13.918 | <0.001 | |
| | While watching reality travel variety programs, I feel as though I am in the village. | 0.842 | 0.077 | 15.547 | <0.001 | |
| | While watching reality travel variety programs, I feel like a resident of the village. | 0.854 | 0.081 | 15.765 | <0.001 | |
| | While watching reality travel variety programs, I feel like I am traveling. | 0.840 | 0.082 | 15.513 | <0.001 | |
| Entertainment | Reality travel variety programs are more entertaining than other reality variety programs. | 0.701 | | | | 0.846 |
| | Viewing reality travel variety programs makes me happy. | 0.824 | 0.066 | 14.348 | <0.001 | |
| | Every new episode of a reality travel variety program is interesting. | 0.873 | 0.069 | 15.061 | <0.001 | |
| | Places that appear in reality travel variety programs are more interesting than in other reality variety programs. | 0.780 | 0.075 | 13.637 | <0.001 | |

**Table 2.** *Cont.*

| Construct | Measurement Items | Factor Loading | S.E. | C.R. | *p*-Value | Cronbach's a |
|---|---|---|---|---|---|---|
| Information-seeking | Reality travel variety programs show specific information about tourism destinations. | 0.758 | | | | |
| | Reality travel variety programs show cultural information about tourism destinations. | 0.652 | 0.068 | 12.254 | <0.001 | |
| | Reality travel variety programs are educational in showing the way of life in tourist destinations. | 0.776 | 0.073 | 14.844 | <0.001 | 0.863 |
| | Reality travel variety programs are beneficial because they provide an overview of tourism destinations. | 0.772 | 0.075 | 14.755 | <0.001 | |
| | Reality travel variety programs have enhanced my intellectual ability to travel by making it possible to know about tourism destinations. | 0.779 | 0.068 | 14.892 | <0.001 | |
| Habitual time-spending | Reality travel variety programs are good for spending time alone. | 0.820 | | | | |
| | Reality travel variety programs are good to watch without thinking. | 0.827 | 0.056 | 18.046 | <0.001 | 0.899 |
| | I watch reality travel variety programs habitually without a special purpose. | 0.814 | 0.059 | 17.658 | <0.001 | |
| | There are no other programs to watch at the time this program is broadcast. | 0.863 | 0.056 | 19.176 | <0.001 | |
| Socializing | I watch reality travel variety programs to empathize with others. | 0.642 | | | | |
| | Reality travel variety program are subjects of interest among people. | 0.557 | 0.134 | 8.753 | <0.001 | 0.746 |
| | I watch reality travel variety programs to avoid being alienated from conversations with others. | 0.745 | 0.144 | 9.487 | <0.001 | |
| Viewing satisfaction | I am satisfied with the content of reality travel variety programs. | 0.650 | | | | |
| | I would like to watch more reality travel variety programs. | 0.783 | 0.096 | 12.844 | <0.001 | 0.803 |
| | I am satisfied that I have gained useful information about places and cultures that appear in reality travel variety programs. | 0.911 | 0.101 | 14.400 | <0.001 | |
| | I would like my acquaintances to recommend watching reality travel variety programs. | 0.907 | 0.103 | 14.349 | <0.001 | |
| Presence | I feel that I am in the TV situation with the characters. | 0.709 | | | | |
| | I feel that the situations on TV are happening in front of my eyes. | 0.815 | 0.066 | 14.020 | <0.001 | 0.887 |
| | The characters on the TV seem to be talking right in front of me. | 0.890 | 0.076 | 14.640 | <0.001 | |
| | I feel that I am in that place on the TV. | 0.645 | 0.073 | 11.292 | <0.001 | |
| Attitude toward tourism destinations | I feel that the places that appear on reality travel variety programs are good. | 0.759 | | | | |
| | I feel that the places that appear on reality travel variety programs are good places to travel. | 0.769 | 0.068 | 14.924 | <0.001 | 0.882 |
| | I like the places that appear on reality travel variety programs. | 0.847 | 0.068 | 16.637 | <0.001 | |
| | I feel that the places that appear on reality travel variety programs are attractive. | 0.854 | 0.067 | 16.806 | <0.001 | |
| $X^2$ = 982.850, *df* = 467, *p* = 0.000, (*AGFI* = 0.909 *TLI* = 0.924, *CFI-* = 0.933, *RMSEA* = 0.056) | | | | | | |

**Table 3.** Correlation analysis of constructs and average variance extracted (AVE).

| | Vicarious Gratification | Entertainment | Information-Seeking | Habitual Time-Spending | Socializing | Viewing Satisfaction | Presence | Attitude toward Tourism Destination |
|---|---|---|---|---|---|---|---|---|
| Vicarious Gratification | (0.636) | | | | | | | |
| Entertainment | 0.458 ** | (0.661) | | | | | | |
| Information-Seeking | 0.457 ** | 0.560 ** | (0.701) | | | | | |
| Habitual Time-Spending | 0.388 ** | 0.536 ** | 0.722 ** | (0.759) | | | | |
| Socializing | 0.452 ** | 0.469 ** | 0.491 ** | 0.364 ** | (0.711) | | | |
| Satisfaction | 0.464 ** | 0.528 ** | 0.588 ** | 0.521 ** | 0.593 ** | (0.745) | | |
| Presence | −0.029 | 0.003 | −0.020 | 0.020 | 0.012 | 0.049 | (0.749) | |
| Attitude toward Tourism Destination | 0.417 ** | 0.655 ** | 0.576 ** | 0.538 ** | 0.483 ** | 0.744 ** | 0.043 | (0.758) |

Note: Values in parentheses are AVE, and the lower values of the diagonal indicate the factor correlation: $* p < 0.05$, $** p < 0.01$.

### 4.3. Verification of Hypotheses

Given an acceptable measurement-model fit, SEM was carried out in AMOS to investigate the overall fit of the structural model and hypotheses. The fit of the proposed model was tested using fit indices. The results ($\chi^2$/df = 958.581/483, CMIN/df = 1.984, RMR = 0.028, GFI = 0.900, AGFI = 0.902, CFI = 0.908, NFI = 0.901, IFI = 0.909, RMSEA = 0.063) proved that the model was a good fit and appropriate for the sample data. Therefore, the analysis was performed without any model modification. The results of the hypotheses in this study are summarized in Table 4.

**Table 4.** Results of hypothesis testing.

| | Structural Paths | β | t-Value | Hypothesis Test |
|---|---|---|---|---|
| H1-1 | Vicarious Gratification → Presence | 0.453 | 4.899 *** | Supported |
| H1-2 | Entertainment → Presence | 0.286 | 4.030 *** | Supported |
| H1-3 | Information-Seeking → Presence | 0.131 | 2.182 * | Supported |
| H1-4 | Habitual Time-Spending → Presence | 0.076 | 1.302 | Rejected |
| H1-5 | Socializing → Presence | 0.129 | 1.937 | Rejected |
| H2 | Presence → Viewing Satisfaction | 0.239 | 4.517 *** | Supported |
| H3 | Viewing Satisfaction → Attitude toward Tourism Destination | 0.818 | 10.630 *** | Supported |
| Model fit | Chi-square/df = 958.581/483, CMIN/df = 1.984, RMR = 0.028 GFI = 0.900, AGFI = 0.902, CFI = 0.908, NFI = 0.901, IFI = 0.909, RMSEA = 0.063 | | | |

Note:$* p < 0.05$, $** p < 0.01$, $*** p < 0.001$.

Specifically, in terms of Hypothesis 1, *Viewing motivation of reality travel variety programs positively affects presence*, Hypothesis 1-1, which assumes that vicarious gratification positively affects presence, was positive and significant at the 5% level. Hypothesis 1-2, which assumes that entertainment positively affects presence, was positively related to presence and significant at the 5% level. Hypothesis 1-3, which assumes that information-seeking positively affects presence, was also positive and significant at the 5% level. Hypotheses 1-4 and 1-5, which posit that habitual time-spending and socializing positively affect presence, were not significant and not supported. Regarding Hypothesis 2, which assumes that presence positively affects viewing satisfaction of reality travel variety programs, was positive and significant at the 5% level, thus supporting this hypothesis. Hypothesis 3, which assumes

that viewing satisfaction of reality travel variety programs positively affects attitudes toward tourism destinations, was positive and significant at the 5% level, supporting this hypothesis as well.

## 5. Discussion and Conclusions

### 5.1. Discussion

This study was conducted with viewers who have experience watching travel reality variety programs; the program format is travel reality show, which is a mixture of travel and reality rather than a movie or drama. This aspect seems to contribute in a way that is different from previous research. In addition, previous studies investigated only the relationship between motivation for viewing a reality program and satisfaction in various aspects, but few studies have analyzed the psychological effect on viewing motivation and satisfaction by presence, by which viewers vividly and realistically recognize the places in the programs. Thus, it can be considered that the attempt to identify presence in travel reality variety programs is a contribution to the scalability of the theoretical application utilized in this study. The results of this study are as follows.

First, among five motivational factors, vicarious gratification, entertainment, and information-seeking have a positive influence on presence. This is because of the nature of travel reality variety programs, which provide destination information and entertainment at the same time, provide accurate information on the locations, then have fun with the content itself, and the vicarious gratification provided by the indirect experience with this kind of program gives viewers a vivid sense of aliveness or presence about the place, which leads to tourists expectations toward the place. This result is meaningful, because it proves that the effect of presence is mediated in travel reality variety programs as well as reality programs [10,48]. Second, it was confirmed that presence had a positive effect on the satisfaction of viewing travel reality variety programs, similar to the presence of reality programs such as "Cookbang" and "Mukbang" [15,24]. It can be concluded that viewers can experience more presence in the place due to the liveliness, which is one of the features of travel variety programs, and this presence has a positive influence on viewing satisfaction. Therefore, presence can be said to be a significant parameter that can influence the relationship between viewing motivation and satisfaction. Third, a positive attitude toward the tourism destination in the program was found when the viewing satisfaction was higher in this relationship. This supports previous research on satisfaction and attitude [49]. Ultimately, in travel reality variety programs, like other reality programs, viewing satisfaction causes the viewers to have good image and expectation for the place, and finally, viewing satisfaction can lead to a positive attitude toward tourism destinations [50].

### 5.2. Implications

Above all, we note the theoretical implications of this study. First, this study divided viewing motivation into five variables based on a literature study on previous media-viewing motivation, and it is meaningful that this research considered media diversity by applying reality travel variety programs. In particular, among five motivational factors, considering the characteristics of the travel reality program, adding the variable of "vicarious gratification" as an indirect experience through the screen can also be said to be a meaningful theoretical extension from the previous research.

In addition, presence, the realistic feeling given by a reality travel variety program, was verified empirically as a mediating role between viewing motivation and viewing satisfaction. This is an attractive factor for viewers watching reality programs, so it can be applied to various genres of media in the future. This is also an empirical evidence for the claim that presence is applicable to all media, although there are differences in the type and exposure degree of media [12], and it was confirmed that the experience of presence is a highly significant variable in the travel reality program.

Lastly, it is meaningful that we reconfirmed (based on existing research) the satisfaction of viewing these TV programs and attitudes toward their tourism destinations.

In addition, this study suggests the following practical implications. People often indirectly experience tourism destinations through mass media without actually visiting them. This indirect experience will have a great influence on future behavioral intentions [12,13]. In Korea, due to the diversification of broadcasting channels, each broadcasting company produces programs that include indirect advertisements. Indeed, since the tvN "Over Flowers Series" travel program was broadcast, the number of Korean tourists who visited the broadcast country increased by an average of 52.7% [51]. In this respect, therefore, the results of this study can provide a guideline for mapping strategies to attract tourists to certain places in terms of marketing the tourism destination itself. In particular, the exposure of places in travel reality programs is more realistic and specific than in dramas or movies. In the end, this evokes expectations of a tourism destination [16], and since it approaches the viewer with easy and comfortable images, the effect of enhancing the attractiveness of the space will be strong. Therefore, tourism destinations exposed through travel reality programs are imprinted with a positive image for viewers or potential tourists, and can also affect actual behaviors such as visiting certain locations.

### 5.3. Limitations and Future Research

Finally, we present the limitations of this study and the direction of future research. In this study, even though the questionnaire survey was conducted for respondents who had watched the programs, their memories may have been distorted over time. Therefore, subsequent research should be conducted on programs that are being broadcast to obtain more accurate results. In addition, recent travel reality variety programs may also include entertainers of specific genders or ages. In further studies, it will be necessary to consider the ages and genders of the entertainers on various programs. Lastly, the sample for the survey was limited to adults residing in Busan and Gyeongsangnam provinces. Therefore, further research should consider all regions in Korea. In future research, it is expected that more specific and diverse studies on travel reality programs will be conducted based on this study. The results of this study can then be supplemented with meaningful facts, which will enable a wider understanding of the results and generalization of previously discovered theories.

**Author Contributions:** Conceptualization, B.-K.K.; methodology, K.-O.K.; data curation, K.-O.K.; writing—original draft preparation, B.-K.K.; writing—review and editing, B.-K.K.; project administration, B.-K.K.; funding acquisition, K.-O.K. All authors have read and agreed to the published version of the manuscript.

**Funding:** This work was supported by the Ministry of Education of the Republic of Korea and the National Research Foundation of Korea (NRFS1A5B5A07XXXXXX).

**Conflicts of Interest:** The authors declare no conflict of interest.

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
