# Peer review of "Relationship between Viewing Motivation, Presence, Viewing Satisfaction, and Attitude toward Tourism Destinations Based on TV Travel Reality Variety Programs"

_sustainability, doi:10.3390/su12114614_

Round 1

Reviewer 1 Report

Comments:

This study examined the relationship between the viewing motivation for real variety programs and the viewing experience (presence) during watching, viewing satisfaction after watching, and the attitude toward presented tourism destinations.

The author(s) suggested a conceptual model in the tourism industry and tested whether the proposed relationships were significant using SEM.

The major weakness of this research is its justification for the proposed hypotheses, weak literature review, poor writing, and conclusion. The overall quality of this paper is weak in terms of advancement of theory, literature review, and unique implications for the practitioners. Thus, this reviewer feels that the contribution of current study is questionable. The followings are specific comments: 

First of all, writing is very poor. In many places, this reviewer had a huge difficulty in understanding the author(s) intentions. The author(s) should practice writing in English. Or, the author(s) should send the manuscript for proofreading to the native English speaker before submission.

This research should find a stronger argument regarding (1) why the study is important and (2) why the findings are meaningful for scholars/practitioners.  Please use google scholar, and search the relevant publications. The author(s) can find many similar studies. For this reason, this reviewer thinks that this research does not provide publishable new/fresh information for the readers.

The framing of the work needs a broader conceptual motivation. For instance, the literature review was scattered and mostly offered a list of precedents in the literature to justify the choices of constructs/variables. Instead, in your introduction, focus on building a conceptual

framework and make it clear to the reader what theoretical points are at stake in your research.

How the author(s) check ‘non-respondent error- check’? There is no mention about the important issue in this manuscript.

SEM is widely used in social science studies. One thing that we have to understand is that SEM is a tool which should test a theoretical model. If the model is not theoretically solid, the results could be data-driven. In this respect, what the author explained for potential links between the proposed constructs is not solid enough. Overall, the authors' justification for the hypothesis development requires further explanations. Especially, more support (i.e., theoretical background and empirical evidences) needs to be provided in relation to hypothesis H1-2, H1-3, H1-5, and H2. All the hypotheses should be developed based on thorough literature review…

Conclusion: The discussion is a little disappointing and could be written in a much stronger way. Also, there are very little managerial implications, and more specific, poignant recommendations should be provided to the practitioners/managers. This information would give the paper a much better finish.

In addition, the variables and constructs used are not interesting

There is no answer to the "so what" question

There is nothing here to inspire future research or implications for practice

The model and relationships are less than compelling

The results are clear, but nothing new. The findings do not provide useful guidelines for scholars/practitioners.

Manuscript preparation: The author(s) should follow the submission guideline for Sustainability. Especially, REFERENCE does not follow the submission guideline for Sustainability.

Author Response

Based on the commnets you gave, we revised and supplemented the article with all our heart.

Your comments and in-depth opinions gave us an opportunity to figure out a lot about the shortcomings of this article and what part of the article to pay attention to in the fufure research.

It is an honor and once again, we appreciate your comments.

Reviewer 2 Report

Thank you for a possibility to review such interesting paper.

The article is well written and well structured, logic, but demands a few improvements:

  1. In my opinion using “Wikipedia” as a scientific basis for framework development is rather improper, especially from the perspective attention-grabbing behaviour phenomenon in modern media. There are many papers focused on such topic.
  2. After reading the text, I stated that the secondary aim of the paper, i.e. developing of specific tourist expectations (TE) towards destinations of TV tourist shows viewers could be more added. In my opinion, the phenomenon of TE should be included in literature review section due to its potential impact on further watchers behaviours as a consumer/tourist. The content of TV shows and its attributes could be a trigger in decision making process and possess several practical implications.
  3. Generally, I have no remarks for methodology and presentation of results. In my subjective opinion sections Research Model and Methodology as well as Results are well written, but the Data Collection and Analysis Method subsection should be expanded with a more complete description of the statistical methods used, including scientific sources.
  4. On Discussion section the aspects concerning potential tourist expectations formation could be added - especially from marketing perspective and practical implications of results.
  5. Implications subsection. The practical implications of the results from the perspective of TE development should be more detailly described.

Author Response

(The authors gave the same response as above.)

Reviewer 3 Report

The paper reports on an interesting research however in its current form it is unsuitable for publication.

Many sentences are badly written making them unclear, e.g. (the numbers in quotes are row numbers):

In particular, since tourism destinations in travel real variety programs are likely to get far more 51 exposure than in movies or dramas, and the tourism destination itself , which is the setting for the 52 broadcast program, is a major part of the broadcasting theme with rational appeal, it can be 53 understood that viewers’ involvement in the tourism destination has increased, motivating viewers 54 to travel (Cho & Shin, 2016).

(chaotic, hard to decipher what is the argument)

Koreans are 56 mostly influenced by TV programs in the Asia-Pacific region

(unclear what is the argument, are the Koreans the most influenced in the Asia-Pacific region compared to viewers from compared to other nations, or the Koreans are most influenced by TV programs set in the Asia-Pacific region compared to TV programs set in other regions)

The research goal is stated twice (rows 66-71 and 136-139), the latter in a less clear way.

The "Data Collection and Analysis Method" section does not provide any information on how the survey participants were recruited. Were they chosen randomly (from what population?) or was it a convenience sample (again, based on what population)? If it was the latter, it constitutes a significant limitation and it should be described in section 5.4.Limitations and future research.

The discussion is also unclear, e.g.:

316 However, socializing and habitual time-spending were rejected. This is supported by the fact 317 that the viewing motivation may be different, depending on the type of media from the previous 318 studies.

(chaotic, concepts are mixed with hypotheses)

This result can be said that because of the live nature, which is one of the 323 characteristics of real travel variety program (Fu et al., 2016), presence and sense of reality, that 324 viewers feel can affect viewing satisfaction.

(what is the point here?)

The authors mix their own conclusions with other researchers':

338 In addition, this study suggests the following practical implications. In fact, an article based on 339 the statistics of “Sky Scanner,” a travel specialist site in 2019, analyzes TV entertainment and 340 travel-market trends.

- The authors should compare their results with other researchers' but it should be clear what is their own result and what is not.

I do not know whether these unclear sentences are all a result of chaotic writing, or some of them are simply badly translated from Korean, therefore I suggest consulting the text with a professional translator.

Technical remarks:

The references should be separated with spaces from the main text, and they are not, e.g.:
123 which is only auditory(Bracken, 2009).
128 differ depending on the reality program sub-type(Kim & Lee, 2014).
131 information-seeking) and viewing satisfaction(Jang & Kim, 2016).

Figure 1 is of very low resolution.

The content of Table 4.1.Profile of the Sample is to much extent duplicated in its description in the main text. The description should provide a comment for the table, there is no need to repeat most of its content.

The reference Wikipedia is not only badly specified (it should describe the specific page it points to, not merely the Wikipedia), but the provided link (https://ko.wikipedia.org/wiki/%EB%A6%AC%EC%96%BC_%EB%B2%84%EB%9D%BC%EC%9D%) does not lead to any content (only an error message).

The references do not have DOIs provided.

Author Response

(The authors gave the same response as above.)

Round 2

Reviewer 1 Report

There is improvement in the revised manuscript and the authors have responded satisfactorily to the original concerns raised by the reviewers. The authors provided more thorough literature reviews and stronger justification for proposed research framework.

However, conclusion section is still weak.

It is necessary to provide a lot stronger theoretical/practical implications.

Author Response

Thanks for the in-depth comments until the end.
Thanks to you, we are very grateful to be able to submit a high-quality paper.
